# 3D Medical Image Segmentation with Anatomy-Guided Conditioning from Surrounding Structures

## Abstract

Accurate segmentation of complex anatomical structures in 3D medical images is challenged by low contrast, unclear features, and complex topology. We propose Anatomy-Guided Conditioning (AGC), which integrates signed distance maps of surrounding anatomy into segmentation networks via feature modulation in the decoder. Anatomical priors are obtained from automatic tools such as TotalSegmentator, requiring no additional training and enabling use in both multi-class and single-target tasks. We evaluate AGC on CTA coronary arteries, PET/CT visceral fat, CT head-and-neck organs, and CBCT dental canals. Across CNN, Transformer, and hybrid backbones, AGC improves Dice, HD95, and topology-aware metrics (clDice, Betti error), reducing boundary errors and fragmentation. These results demonstrate that conditioning on surrounding anatomy provides a simple and broadly applicable inductive bias for anatomically constrained 3D segmentation.

## 1 Introduction

Deep learning has achieved remarkable success in 3D medical image segmentation, with models reaching expert-level accuracy on benchmarks such as MSD (Antonelli et al., 2022) and BraTS (Menze et al., 2015; Bakas et al., 2018). Yet most methods still rely heavily on local image appearance and voxel-wise objectives, often neglecting global anatomical context. This limitation leads to anatomically implausible predictions, including fragmented vessels, leakage beyond organ boundaries, and topological inconsistencies (Wyburd et al., 2024; Bassi et al., 2024). Such problems are particularly severe in coronary artery segmentation, where continuity must be preserved through low-contrast and artifact-prone regions (Duan et al., 2019).

Radiologists routinely delineate structures by referencing surrounding anatomy rather than considering each organ in isolation. This observation motivates the need for segmentation frameworks that explicitly incorporate anatomical context from anchor anatomical regions.

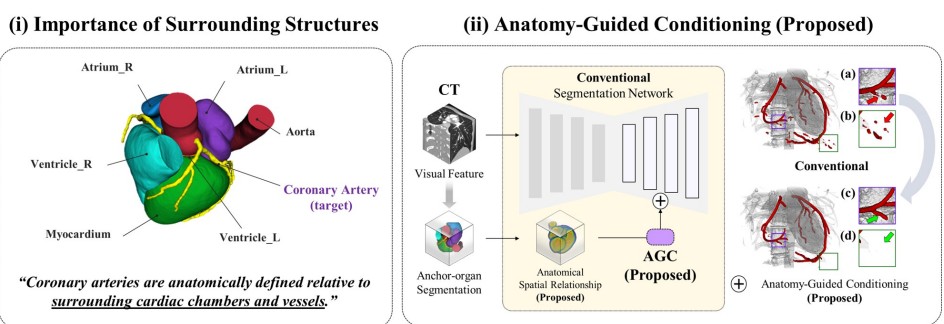

Figure 1: Illustration of coronary artery segmentation. Left: baseline prediction with fragmented branches and boundary leakage. Right: our proposed AGC reduces errors by leveraging surrounding anatomical context.

We introduce **Anatomy-Guided Conditioning (AGC)**, a simple conditioning module that integrates signed distance maps of neighboring structures into segmentation networks. Unlike binary mask or coordinate priors Guo et al. (2020); Goncharov et al. (2024), distance maps encode continuous proximity information, enabling more precise spatial reasoning. By injecting these priors at the feature level via FiLM-based modulation Perez et al. (2018), AGC enhances structural coherence without altering inputs or encoders.

We evaluate AGC on diverse datasets, including thin tubular vessels (ImageCAS), diffuse abdominal regions (VAT PET/CT), small head-and-neck organs (HAN-Seg), and narrow dental canals (Tooth-Fairy2). These benchmarks present complementary challenges such as continuity, boundary precision, and variability across both CT and CBCT modalities. AGC consistently improves overlap, boundary, and topology-aware metrics, and provides complementary benefits when combined with recent topology-preserving losses such as clDice Menten et al. (2023) and Skeleton Recall Kirchhoff et al. (2024).

Our contributions are threefold:

- We propose **AGC**, a conditioning mechanism based on signed distance maps of surrounding anatomy, enabling proximity-aware reasoning beyond categorical masks or coordinates.

- We demonstrate that AGC consistently improves structure-aware metrics and acts synergistically with topology-preserving objectives, enhancing continuity and reducing topological errors.

- We show that AGC is **simple, backbone- and dataset-agnostic**, requiring no manual annotation by leveraging off-the-shelf tools such as TotalSegmentator Wasserthal et al. (2023).

## 2 RELATED WORKS

**Network Design for Medical Image Segmentation.** Volumetric segmentation has been explored with a broad range of architectures, from convolutional encoder–decoders (e.g., 3D U-Net (Çiçek et al., 2016), V-Net (Milletari et al., 2016), SegResNet (Myronenko, 2019)) to transformer-based and hybrid designs (e.g., TransUNet (Chen et al., 2024), UNETR (Hatamizadeh et al., 2022), Swin-UNETR (Hatamizadeh et al., 2021)). Large-scale benchmarks such as nnU-Net (Isensee et al., 2021) and Touchstone (Bassi et al., 2024) show that performance is often task-dependent, and that carefully tuned CNNs can match more complex backbones Azad et al. (2024). However, architectural refinements alone are insufficient for thin or low-contrast structures, where local features are unreliable. These cases motivate the use of anatomical priors that are model-agnostic and can provide organ-relative context beyond raw image intensities.

**TotalSegmentator as a Source of Anatomical Priors.** *TotalSegmentator* (Wasserthal et al., 2023) is a foundation model trained on over 1,200 CT scans that outputs robust pseudo-labels for 104 organs, bones, and vessels. Its coverage and generalization make it a scalable source of anatomical context, readily applicable to CT, CBCT, and even MRI. Importantly, these pseudo-labels require no manual annotation, enabling downstream tasks to access surrounding anatomy without extra supervision. This capability allows priors to be introduced even in single-target problems such as coronary artery segmentation, where auxiliary multi-class labels are typically unavailable. In our framework, this factor is a key enabler of broad applicability.

**Anatomical Prior-based Conditioning.** Several strategies have been proposed to incorporate anatomical knowledge. Topology-aware losses such as clDice (Shit et al., 2021) and Skeleton Recall (Kirchhoff et al., 2024) encourage connectivity via differentiable skeletonization, but remain confined to the optimization stage and require task-specific tuning. Shape-based approaches, e.g., SPGNet (Song et al., 2024), depend on explicit shape models learned from annotated datasets, limiting scalability to diverse anatomies or 3D volumes. Coordinate embeddings (Das et al., 2024) add absolute voxel positions but lack anatomical semantics, reducing generalizability across architectures. Anchor-based methods (Guo et al., 2020) sequentially segment organs using binary masks as anchors; while effective in multi-target head-and-neck settings, they are unsuitable for single-target tasks and cannot encode continuous distance information.

## 3 METHODS

### 3.1 PROPOSED FRAMEWORK

AGC enhances segmentation by embedding spatial context from surrounding anatomy into the decoding process. Surrounding structures are obtained from off-the-shelf TotalSegmentator, which produces pseudo-labels for over 100 organs across CT, CBCT, and MRI without requiring manual annotation. These pseudo-labels are converted into signed distance maps to provide continuous proximity information, which is then injected via FiLM-based feature modulation in the decoder (Figure 2). This design distinguishes AGC from prior mask- or shape-based conditioning, offering a simple, backbone-agnostic mechanism that supplies proximity-aware guidance even in single-target tasks such as coronary artery segmentation.

**(a) Overall Network Architecture**

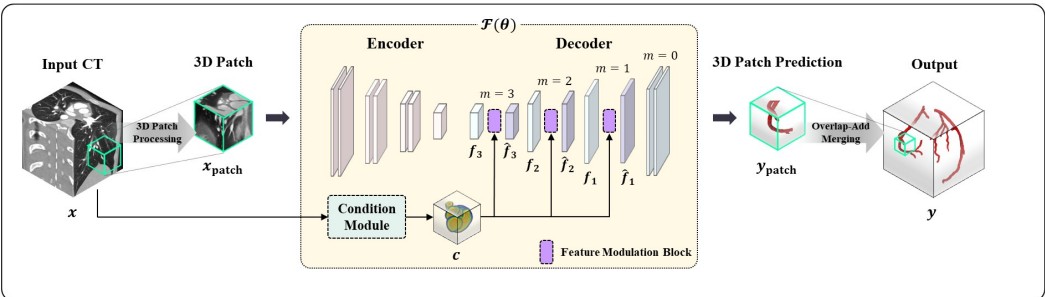

**(b) Condition Module**

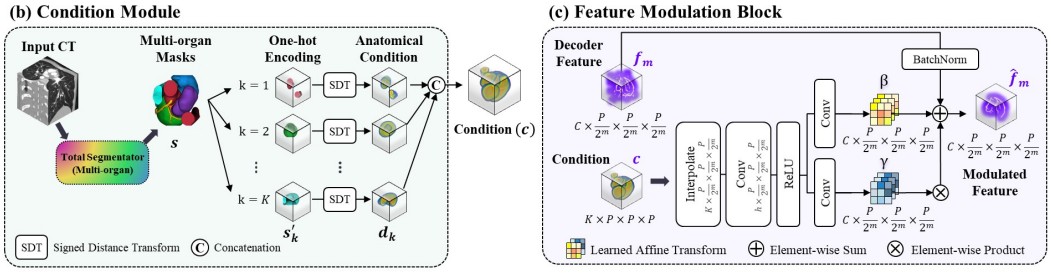

Figure 2: Architecture of the proposed Anatomy-Guided Conditioning (AGC) framework, including the overall network (a), condition module (b), and feature modulation block (c).

**Network Architecture**  The proposed **AGC** method is integrated into a standard encoder-decoder architecture and follows a patch-based framework commonly adopted in 3D medical image segmentation Isensee et al. (2021). To support clarity, we briefly outline the principle of general patch-based segmentation framework.

As illustrated in Figure 2 (a), a full CT volume $x \in \mathbb{R}^{1 \times H \times W \times D}$ is divided into overlapping 3D patches $x_{\text{patch}} \in \mathbb{R}^{1 \times P \times P \times P}$, where $P = 96$ in our experiments. Each patch is processed independently, and the resulting predictions are aggregated into a full-resolution output $y \in \mathbb{R}^{2 \times H \times W \times D}$ using a weighted averaging scheme with a 3D Gaussian kernel and 25% overlap, mitigating boundary artifacts at patch edges. As seen in the notation above, binary segmentation tasks are formulated as two-class problems (foreground and background) following standard practice to maintain stability and compatibility with conventional loss functions with multi-class based cross-entropy. The following sections detail how anatomical priors are extracted and encoded, and how they modulate decoding features via our feature modulation mechanism.

**Anatomical Condition Module**  We extract anatomical priors by first applying a multi-organ segmentation tool, such as **TotalSegmentator** (Wasserthal et al., 2023), to an input CT volume $x \in \mathbb{R}^{H \times W \times D}$. This model produces a semantic pseudo label map $s$:

$$s \in \{1, 2, \ldots, K\}^{H \times W \times D},$$

where each voxel is assigned an integer label corresponding to one of $K$ anatomical anchor structures. We convert this label map into a one-hot encoded tensor $s'$:

$$s' \in \{0, 1\}^{K \times H \times W \times D},$$

where the $k$-th slice $s'_k$ is a binary mask indicating the presence of organ $k$. The region of organ $k$ is defined as:

$$\mathcal{O}_k = \{\mathbf{p} \in \Omega \mid s'_k(\mathbf{p}) = 1\},$$

where $\Omega = \{1, \ldots, H\} \times \{1, \ldots, W\} \times \{1, \ldots, D\}$ is the full spatial domain.

To represent soft spatial context beyond hard anatomical boundaries, we compute a **signed distance transform** for each organ $k$, resulting in a signed distance map $d_k \in \mathbb{R}^{H \times W \times D}$ defined as:

$$d_k(\mathbf{p}) = \begin{cases} - \min\limits_{\mathbf{q} \in \Omega \setminus \mathcal{O}_k} \|\mathbf{p} - \mathbf{q}\|_2, & \text{if } \mathbf{p} \in \mathcal{O}_k, \\ \min\limits_{\mathbf{q} \in \mathcal{O}_k} \|\mathbf{p} - \mathbf{q}\|_2, & \text{if } \mathbf{p} \notin \mathcal{O}_k, \end{cases}$$

where $\mathbf{p}, \mathbf{q} \in \Omega$ denote voxel coordinates and $\| \cdot \|_2$ is the Euclidean norm. This results in negative values inside the organ ($\mathbf{p} \in \mathcal{O}_k$), positive values outside the organ ($\mathbf{p} \notin \mathcal{O}_k$), and a zero level set at the organ boundary. Finally, we stack all organ-wise distance maps into an anatomical prior tensor $c$:

$$c = \{d_k\}_{k=1}^{K} \in \mathbb{R}^{K \times H \times W \times D},$$

The anatomical prior $c$ serves as a rich spatial prior that captures both the absolute and relative anatomical positioning. It is extracted as $K \times P \times P \times P$ 3D patches and used as conditioning input for downstream components (see Figures 2(a) and (b)).

**Anatomy-Guided Feature Modulation** To integrate anatomical context with minimal interference, we inject the generated priors into the decoder using a Feature Modulation Block (see Figure 2(c)). In particular, many existing methods leverage pre-trained encoders on large-scale datasets, which produce features where general-purpose priors are well-preserved. Modifying the encoder can disrupt these valuable features and degrade performance. Therefore, we focus on integrating the generated anatomical priors into the decoder. This design allows anatomical priors to guide feature refinement at later stages, where semantic information is richer, without altering early feature extraction.

At each decoder stage $m$, let $f_m \in \mathbb{R}^{C \times \frac{P}{2^m} \times \frac{P}{2^m} \times \frac{P}{2^m}}$ be the decoder feature map, and let $c$ be the anatomical prior tensor. We first apply trilinear interpolation to $c$ to match the spatial dimensions of $f_m$, resulting in a down-sampled prior of shape $K \times \frac{P}{2^m} \times \frac{P}{2^m} \times \frac{P}{2^m}$. The interpolated prior is then processed by a lightweight convolutional block to extract modulation parameters. Specifically, a $3 \times 3 \times 3$ convolution first projects the $K$-channel prior into a higher-dimensional hidden representation (128 channels), followed by a ReLU activation. This intermediate representation is then passed through two parallel $3 \times 3 \times 3$ convolutions to produce the spatially adaptive scaling and shifting parameters, $\gamma$ and $\beta$, respectively, both in $\mathbb{R}^{C \times \frac{P}{2^m} \times \frac{P}{2^m} \times \frac{P}{2^m}}$. These modulation parameters are applied to the normalized decoder features via FiLM-style conditioning Perez et al. (2018):

$$\hat{f}_m = \gamma \odot \text{BN}(f_m) + \beta, \tag{1}$$

where $\text{BN}(\cdot)$ denotes batch normalization and $\odot$ is element-wise multiplication. This modulation allows the network to adjust decoder features in a spatially adaptive yet minimally invasive manner, correcting anatomical inconsistencies while preserving the underlying feature representation.

## 3.2 DATASETS

We benchmark AGC on four CT/Cone-Beam CT-based datasets that span diverse anatomical targets and challenges: i) ImageCAS for thin, highly branched coronary vessels, ii) VAT PET/CT for spatially diffuse abdominal fat, iii) HAN-Seg for small, heterogeneous head & neck structures under data scarcity, and iv) Tooth-Fairy2 for narrow dental canals. Anatomical masks used for conditioning were either provided with the datasets or generated using off-the-shelf **TotalSegmentator v2.6.0**. Here we summarize only the keypoints and illustrative details of conditioning for each dataset and preprocessing schemes are further demonstrated in **Supplementary Figure 1**.

**ImageCAS—Coronary Arteries** (Zeng et al., 2023). 1,000 contrast-enhanced chest CTA volumes (700/50/250 train/val/test). Target: coronary arteries, prone to centerline fragmentation due to their thin and densely branched morphology. Conditioning: aorta, myocardium, atria, and ventricles from TotalSegmentator.

**Low-dose PET/CT VAT—Visceral Adipose Tissue**. 147 low-dose cone-beam CT scans (108/10/29 train/val/test). Target: visceral adipose tissue spanning the abdominal region, difficult to delineate under low-contrast CBCT. Conditioning: body trunk, torso fat, liver, heart, spine, and ribs.

**HAN-Seg—Head & Neck Organs-at-Risk** (Podobnik et al., 2023). 42 CT scans (30/12 train/test). Targets: carotids, spinal cord, mandible, and thyroid, with challenges of discontinuity, high shape variability, and low tissue contrast. Conditioning: dataset-provided labels of targets and neighboring structures.

**Tooth-Fairy2—Inferior Alveolar Canal** (Bolelli et al., 2024). 417 dental CBCT scans (292/25/100 train/val/test). Target: inferior alveolar canal, a narrow, low-contrast tubular structure critical in dental surgery. Conditioning: dataset-provided label of mandible and surrounding tooth masks.

**Implementation Details** All scans were resampled to isotropic spacing and clipped to task-specific Hounsfield windows (e.g., $-150$–$550$ HU for ImageCAS). Intensity values were min–max normalized to $[0, 1]$. We followed MONAI's reference pipeline Cardoso et al. (2022) using $96^3$ voxel patches, with augmentations including random axis flips (10% probability) and intensity jittering of $\pm 5\%$ (50%).

Models were trained with the Adam optimizer for 200 epochs using a starting learning rate of 0.001 decayed via cosine annealing. The Dice–Focal loss was used to balance overlap accuracy and class imbalance. We report four metrics: (1) **Dice coefficient** for region overlap; (2) **95$^{\text{th}}$-percentile Hausdorff distance** (HD95) for boundary error; (3) **Betti number error** (Hu et al., 2019), which quantifies topological mismatches via $\beta_0$ (components) and $\beta_1$ (loops); and (4) **clDice** (Shit et al., 2021), a connectivity-aware variant suited for thin structures. Higher Dice and clDice are better; lower $\text{HD}_{95}$ and Betti error indicate better geometric and topological fidelity.

# 4 EXPERIMENTS AND RESULTS

## 4.1 EXPERIMENTAL RATIONALE

The experiments are designed to evaluate AGC as a simple and general mechanism for embedding anatomical context across models, datasets, and training objectives. We organize the evaluation along four complementary axes.

**Backbone generality.** We first test whether AGC improves performance consistently across diverse segmentation architectures. Since AGC operates at the feature level, it integrates seamlessly with both convolutional and transformer-based models without modifying inputs or encoders.

**Dataset generalization.** Second, we evaluate AGC across four benchmarks spanning distinct anatomies and modalities: thin coronary vessels (ImageCAS, CTA), diffuse abdominal fat (VAT, PET/CT), small head-and-neck organs (HAN-Seg, CT), and narrow dental canals (Tooth-Fairy2, CBCT). These tasks capture complementary challenges such as continuity, boundary precision, and variability under heterogeneous imaging conditions.

**Relation to topology-aware objectives.** Third, we compare AGC with topology-preserving losses such as clDice Menten et al. (2023) and Skeleton Recall Kirchhoff et al. (2024). We hypothesize complementarity: topology-aware losses regularize connectivity during optimization, while AGC guides feature representations with anatomical priors.

**Alternative conditioning.** Finally, we compare AGC with positional encodings (PPE, APE) and mask priors. These provide only coarse spatial cues, whereas AGC encodes continuous proximity information via distance maps, offering a more principled conditioning mechanism.

## 4.2 MODEL-AGNOSTIC EVALUATION

We first assess whether AGC improves segmentation across convolutional, transformer-based, hybrid, and vascular-specialized backbones. Table 1 shows consistent improvements not only in overlap but also in boundary and topology-aware metrics. Convolutional models benefit most, with AGC reducing discontinuities and false positives, while transformers such as UNETR show alleviated fragmentation and restored connectivity. Qualitative results in Figure 3 confirm enhanced structural coherence across architectures. Supplementary Table 1 reports model-agnostic performance on other datasets.

Table 1: Model-agnostic evaluation of AGC across **seven segmentation backbones on the Image-CAS dataset**. AGC denotes whether our anatomy-guided conditioning was used or not

| Backbone | Architecture | AGC | DSC ↑ | HD95 ↓ | clDice ↑ | $\beta_0$ error ↓ | $\beta_1$ error ↓ |
|---|---|---|---|---|---|---|---|
| SegResNet | Convolutional | | 0.77 ± 0.03 | 81.8 ± 29.1 | 0.82 ± 0.08 | 25.2 ± 8.71 | 22.7 ± 8.30 |
| | Convolutional | ✓ | **0.79 ± 0.02** | **36.4 ± 12.0** | **0.86 ± 0.05** | **10.6 ± 3.94** | **10.0 ± 3.99** |
| VNet | Convolutional | | 0.78 ± 0.02 | 66.0 ± 18.9 | 0.84 ± 0.06 | 19.0 ± 6.75 | 17.3 ± 6.34 |
| | Convolutional | ✓ | **0.79 ± 0.05** | **38.2 ± 25.8** | **0.86 ± 0.06** | **12.4 ± 5.21** | **11.5 ± 4.97** |
| Attention-UNet | Convolutional | | 0.79 ± 0.02 | 57.1 ± 17.4 | 0.84 ± 0.06 | 18.4 ± 6.69 | 16.9 ± 6.22 |
| | Convolutional | ✓ | 0.78 ± 0.02 | **31.1 ± 10.9** | **0.86 ± 0.05** | **10.2 ± 4.46** | **9.56 ± 4.26** |
| UNETR | Transformer | | 0.69 ± 0.03 | 162.7 ± 17.5 | 0.60 ± 0.09 | 221.4 ± 50.6 | 208.9 ± 48.2 |
| | Transformer | ✓ | **0.76 ± 0.02** | **47.9 ± 11.7** | **0.74 ± 0.07** | **86.3 ± 24.0** | **81.0 ± 22.4** |
| Swin-UNETR | Hybrid | | 0.79 ± 0.03 | 55.3 ± 16.2 | 0.84 ± 0.06 | 20.6 ± 7.27 | 19.1 ± 6.90 |
| | Hybrid | ✓ | 0.79 ± 0.02 | **43.9 ± 13.0** | 0.84 ± 0.06 | **19.6 ± 6.25** | **18.6 ± 5.76** |
| nnFormer | Hybrid | | 0.72 ± 0.03 | 122.3 ± 24.3 | 0.74 ± 0.08 | 74.8 ± 27.5 | 72.3 ± 26.5 |
| | Hybrid | ✓ | **0.77 ± 0.02** | **33.9 ± 10.8** | **0.82 ± 0.06** | **31.8 ± 15.6** | **29.8 ± 14.7** |
| CS²Net | Vascular-spec. | | 0.72 ± 0.04 | 92.1 ± 20.5 | 0.76 ± 0.08 | 55.1 ± 18.8 | 51.4 ± 18.1 |
| | Vascular-spec. | ✓ | **0.76 ± 0.02** | **40.2 ± 11.2** | **0.83 ± 0.06** | **28.2 ± 10.6** | **26.0 ± 9.85** |

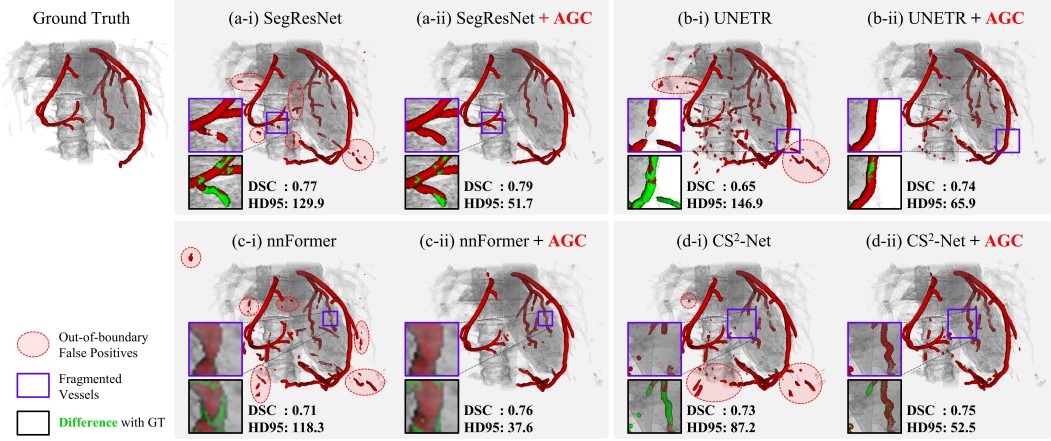

Figure 3: Qualitative 3D visualization of segmentation results on the ImageCAS dataset across multiple segmentation backbone models, with and without Anatomy-Guided Conditioning (AGC).

## 4.3 EVALUATION ON MULTIPLE DATASET

We next evaluate AGC on VAT, HAN-Seg, and Tooth-Fairy2 (Table 2). In VAT, conditioning sharpens diffuse abdominal boundaries. In HAN-Seg, AGC improves challenging small structures such as carotids and mandible under limited data. For Tooth-Fairy2, conditioning on mandible and teeth provides strong anchors for the alveolar canal, yielding highly stable segmentations with minimal topological errors. Qualitative examples are shown in Figure 4, with additional cases in the Supplementary.

Table 2: Comparison between conventional and proposed models on PET/CT VAT, HaN-Seg, and Tooth-Fairy2 datasets. **AGC** column indicates whether our conditioning was used.

| Dataset | AGC | DSC ↑ | mIoU ↑ | HD95 ↓ | $\beta_0$ error ↓ | $\beta_1$ error ↓ |
|---|---|---|---|---|---|---|
| Visceral Adipose Fat (PET-CT VAT) | | 0.94 ± 0.02 | 0.90 ± 0.04 | 1.82 ± 1.55 | 137.4 ± 73.5 | 129.1 ± 70.0 |
| | ✓ | **0.96 ± 0.03** | **0.92 ± 0.04** | **1.56 ± 1.05** | **128.6 ± 48.1** | **119.8 ± 44.6** |
| Carotid Artery (HAN-Seg) | | 0.65 ± 0.07 | 0.48 ± 0.07 | 269.6 ± 74.1 | 36.6 ± 20.2 | 33.3 ± 19.5 |
| | ✓ | **0.73 ± 0.04** | **0.57 ± 0.05** | **19.8 ± 5.15** | **6.25 ± 2.13** | **5.50 ± 1.94** |
| Spinal Cord (HAN-Seg) | | 0.73 ± 0.07 | 0.58 ± 0.08 | 235.0 ± 132.5 | 9.58 ± 6.82 | 8.33 ± 6.17 |
| | ✓ | **0.74 ± 0.05** | **0.59 ± 0.07** | **162.9 ± 27.1** | **5.17 ± 9.35** | **5.08 ± 9.31** |
| Mandible (HAN-Seg) | | 0.84 ± 0.04 | 0.73 ± 0.06 | 261.9 ± 31.1 | 51.3 ± 23.7 | 47.0 ± 20.1 |
| | ✓ | **0.92 ± 0.02** | **0.84 ± 0.03** | **3.06 ± 4.11** | **3.25 ± 6.11** | **2.75 ± 5.07** |
| Thyroid (HAN-Seg) | | 0.78 ± 0.09 | 0.64 ± 0.01 | 215.6 ± 130.0 | 12.6 ± 5.24 | 11.6 ± 4.27 |
| | ✓ | **0.81 ± 0.08** | **0.69 ± 0.11** | **6.57 ± 4.62** | **1.25 ± 0.83** | **1.17 ± 0.80** |
| Alveolar Canal (Tooth-Fairy2) | | 0.89 ± 0.02 | 0.79 ± 0.02 | 2.56 ± 6.89 | 1.82 ± 1.41 | 1.64 ± 1.32 |
| | ✓ | **0.98 ± 0.01** | **0.96 ± 0.02** | **0.68 ± 0.77** | **0.45 ± 0.84** | **0.36 ± 0.68** |

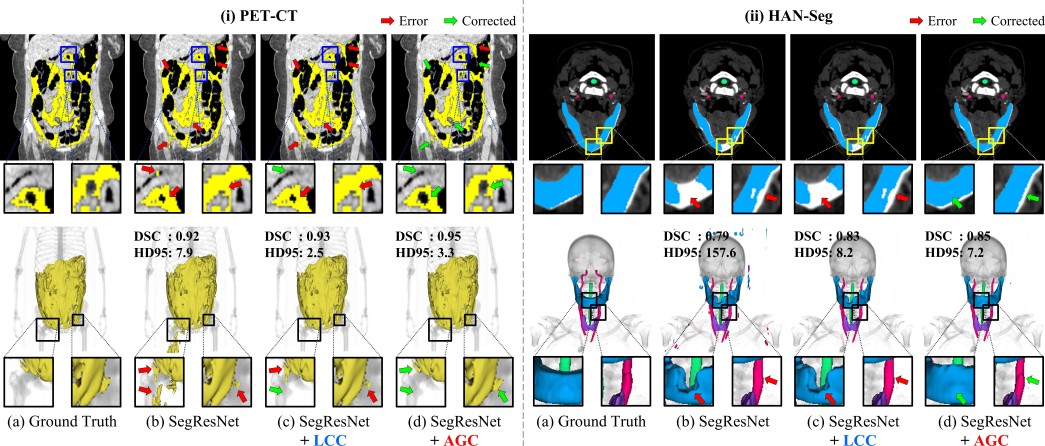

Figure 4: Qualitative comparison of segmentation results on the PET-CT and HAN-Seg datasets. Each column presents the ground truth, SegResNet baseline, and outputs refined by either the largest connected component (LCC) post-processing or our proposed AGC.

## 4.4 LOSS-AGNOSTIC EVALUATION

We next compare AGC with recent topology-aware losses, namely clDice (Menten et al., 2023) (ICCV 2023) and Skeleton Recall (Kirchhoff et al., 2024) (ECCV 2024), both of which enforce 3D centerline and topology preservation during optimization. The comparison addresses two questions: (i) whether AGC performs competitively against these state-of-the-art losses, and (ii) whether they act synergistically when combined. Table 3 shows consistent improvements across losses and datasets, indicating that AGC and topology-aware objectives are complementary: the former injects anatomical context at the feature level, while the latter enforces connectivity during training. The full table, including standard deviations and Betti errors, is provided in Supplementary Table 2.

Metrics such as Dice can sometimes favor DiceFocal despite inferior continuity, reflecting their voxel-wise nature. qualitative figures (Figure 5 clarify such cases, showing that AGC with topology-aware losses reduces fragmentation and improves global coherence even when Dice values are similar.

## 4.5 SPATIAL GUIDANCE ALTERNATIVES

Finally, we compare AGC against other conditioning mechanisms, including physical positional embeddings (PPE), anatomical positional embeddings (APE), and binary mask pri-

Table 3: Loss-agnostic evaluation across datasets. Each subtable reports three different losses (Dice-Focal, SkelRec, clDice) with/without AGC.

### (a) ImageCAS (Coronary)

| Loss | AGC | DSC | HD95 | clDice |
|---|---|---|---|---|
| DiceFocal | | 0.766 | 81.8 | 0.815 |
| | ✓ | **0.788** | **36.4** | **0.862** |
| SkelRec | | 0.777 | 40.6 | 0.856 |
| | ✓ | **0.778** | **39.2** | 0.844 |
| clDice | | 0.786 | 27.9 | 0.855 |
| | ✓ | **0.794** | **27.2** | **0.871** |

### (b) HAN-Seg (Carotid)

| Loss | AGC | DSC | HD95 | clDice |
|---|---|---|---|---|
| DiceFocal | | 0.646 | 269.6 | 0.646 |
| | ✓ | **0.727** | **19.8** | **0.789** |
| SkelRec | | 0.671 | 108.1 | 0.741 |
| | ✓ | **0.698** | 111.5 | **0.796** |
| clDice | | 0.671 | 39.3 | 0.801 |
| | ✓ | **0.673** | **9.28** | **0.841** |

### (c) Tooth-Fairy2 (Alveolar Canal)

| Loss | AGC | DSC | HD95 | clDice |
|---|---|---|---|---|
| DiceFocal | | 0.887 | 2.56 | 0.987 |
| | ✓ | **0.981** | **0.68** | **0.994** |
| SkelRec | | 0.879 | 1.54 | 0.994 |
| | ✓ | **0.988** | **0.36** | **0.997** |
| clDice | | 0.891 | 1.47 | 0.994 |
| | ✓ | **0.989** | **0.31** | **0.997** |

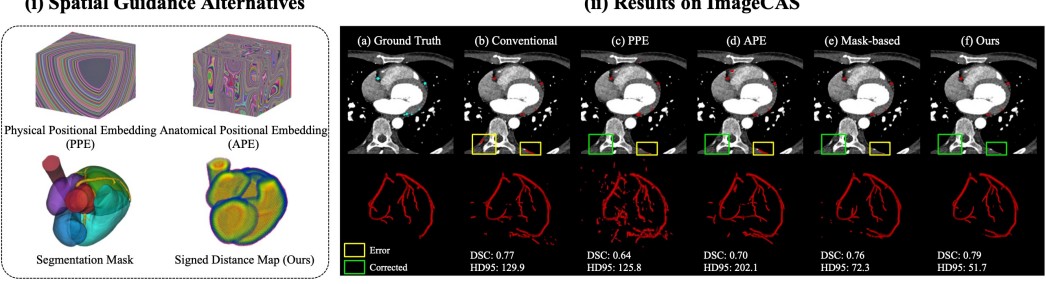

**(i) Coronary Arteries** (ImageCAS)  →Error  →Corrected

**(ii) Carotid Arteries** (HAN-Seg)

**(iii) Alveolar Canal** (Tooth-Fairy2)

(a) Ground Truth  (b) DiceFocal  (c) SkelRec  (d) SkelRec + **AGC**  (e) clDice  (f) clDice + **AGC**

Figure 5: Evaluation across loss functions. Yellow and Green arrow indicate errors in wrong segmentation and where AGC has corrected them

ors. As reported in Table 4, distance map conditioning provides the most consistent improvements. PPE introduces raw coordinates but fails to encode anatomical relevance, while APE shows moderate gains but lacks robustness across tasks.

Mask priors provide categorical boundaries without proximity information, limiting their utility. In contrast, AGC encodes continuous spatial context, resulting in lower topological errors and more coherent predictions. Figure 6 provides qualitative comparisons, confirming that distance maps restore connectivity where alternative priors remain fragmented.

Table 4: Comparison of SegResNet variants using PPE, APE, Mask, and DistMap.

| Method | DSC ↑ | HD95 ↓ | $\beta_0$ ↓ | $\beta_1$ ↓ |
|---|---|---|---|---|
| SegResNet (Base) | 0.77 ± 0.03 | 81.8 ± 29.1 | 25.2 ± 8.71 | 22.7 ± 8.30 |
| + PPE | 0.59 ± 0.03 | 140.2 ± 11.8 | 88.4 ± 23.0 | 78.5 ± 20.4 |
| + APE | 0.77 ± 0.03 | 72.8 ± 22.7 | 19.9 ± 8.10 | 18.6 ± 7.62 |
| + Mask | 0.78 ± 0.02 | 60.3 ± 18.2 | 24.5 ± 7.45 | 22.4 ± 6.99 |
| + DistMap (Ours) | **0.79 ± 0.02** | **36.4 ± 12.0** | **10.6 ± 3.94** | **10.0 ± 3.99** |

**(i) Spatial Guidance Alternatives**

Physical Positional Embedding (PPE)  Anatomical Positional Embedding (APE)

Segmentation Mask  Signed Distance Map (Ours)

**(ii) Results on ImageCAS**

(a) Ground Truth  (b) Conventional  (c) PPE  (d) APE  (e) Mask-based  (f) Ours

□ Error  □ Corrected

DSC: 0.77  DSC: 0.64  DSC: 0.70  DSC: 0.76  DSC: 0.79
HD95: 129.9  HD95: 125.8  HD95: 202.1  HD95: 72.3  HD95: 51.7

Figure 6: Comparison of SegResNet variants using PPE, APE, Mask, and DistMap.

### 4.6 ABLATION STUDY

We conduct two ablation studies to analyze the design choices of AGC. For efficiency and reproducibility, all experiments are performed on a subset of the ImageCAS dataset (210/20/100 train/val/test).

**Encoding Strategy of Distance Map Prior:** A natural concern is whether distance maps actually provide any benefit over simple mask-based priors, since both originate from the same anatomical labels. To address this, we progressively clipped the signed distance maps, starting from a degenerate binary mask ($[0,0]$) and extending the range to include both interior (negative) and exterior (positive) values. As summarized in Table 5a, the binary case yields limited improvements. The full-range variant achieves the best performance overall, demonstrating that distance maps are effective because they encode global spatial positioning across the entire volume.

**Injection Location of the Modulation Block:** The second study examines whether priors should be injected into the encoder, the decoder, or both. This question is particularly relevant when using pretrained encoders such as Swin-UNETR, where modifying early feature representations can be detrimental. Table 5b shows that decoder-only integration achieves the most consistent results, while encoder-only or dual injection provides less stable improvements.

Table 5: Ablation studies: (a) different distance map clipping for anatomical prior injection; (b) injection at different network components.

(a) Distance map clipping strategies

| Threshold | Purpose | DSC $\uparrow$ | HD95 $\downarrow$ | $\beta_0 \downarrow$ | $\beta_1 \downarrow$ |
|---|---|---|---|---|---|
| $[0, 0]$ | Boundary only | $0.76 \pm 0.04$ | $54.1 \pm 25.0$ | $27.5 \pm 6.85$ | $25.3 \pm 6.23$ |
| $[-20, 20]$ | Narrow symmetric | $0.76 \pm 0.04$ | $48.9 \pm 26.0$ | $\mathbf{13.0 \pm 4.43}$ | $\mathbf{12.2 \pm 4.22}$ |
| $[-50, 50]$ | Mid-range symmetric | $0.77 \pm 0.04$ | $47.2 \pm 25.7$ | $20.0 \pm 6.29$ | $18.5 \pm 5.96$ |
| $[-150, 150]$ | Wide symmetric | $0.76 \pm 0.04$ | $52.1 \pm 25.7$ | $18.7 \pm 5.75$ | $17.4 \pm 5.50$ |
| $[\min, \max]$ | Full range | $\mathbf{0.77 \pm 0.04}$ | $\mathbf{36.5 \pm 23.0}$ | $14.3 \pm 5.29$ | $13.3 \pm 4.94$ |
| $[\min, 0]$ | Inside only | $0.75 \pm 0.04$ | $58.9 \pm 23.1$ | $40.8 \pm 10.9$ | $37.0 \pm 9.42$ |
| $[0, \max]$ | Outside only | $0.76 \pm 0.04$ | $49.7 \pm 23.8$ | $20.9 \pm 6.78$ | $19.2 \pm 6.39$ |

(b) Injection depth in the network

| Encoder | Decoder | DSC $\uparrow$ | HD95 $\downarrow$ | $\beta_0 \downarrow$ | $\beta_1 \downarrow$ |
|---|---|---|---|---|---|
| | ✓ | $\mathbf{0.77 \pm 0.04}$ | $\mathbf{36.5 \pm 23.0}$ | $\mathbf{14.3 \pm 5.29}$ | $\mathbf{13.3 \pm 4.94}$ |
| ✓ | | $0.76 \pm 0.04$ | $46.5 \pm 23.1$ | $26.2 \pm 7.94$ | $24.1 \pm 6.95$ |
| ✓ | ✓ | $\mathbf{0.77 \pm 0.04}$ | $41.4 \pm 24.2$ | $25.7 \pm 7.90$ | $23.4 \pm 7.32$ |

## 5 DISCUSSIONS AND CONCLUSIONS

The purpose of this study was not to establish state-of-the-art results on every benchmark, but to examine whether the proposed anatomical guidance can serve as a simple and effective conditioning mechanism in 3D medical image segmentation. Our experiments provide strong evidence that incorporating surrounding anatomy improves both robustness and plausibility. The results on Tooth-Fairy2 are particularly interesting: by conditioning on coarse tooth segmentations, the network was able to delineate the alveolar canal with much improved precision. This illustrates how relative spatial anchors can play a decisive role, and suggests that models can exploit neighbor-aware information in a way reminiscent of human reasoning. More broadly, we observed that AGC consistently reduces false positives appearing in anatomically implausible regions, showing that conditioning provides a direct mechanism for enforcing spatial plausibility that conventional voxel-wise losses cannot guarantee.

While our implementation relies on FiLM-based feature modulation, which acts as a conservative form of error correction, the results indicate that even such a simple strategy is highly effective. This raises the possibility that more expressive conditioning mechanisms could further enrich feature representations while preserving stability. Similarly, our current formulation relies on distance maps from neighboring anchors, capturing relative proximity but not global position. Combining such relative cues with global anatomical encodings, in the spirit of positional embeddings, could provide complementary benefits. Finally, our experiments suggest that conditioning does not require perfect labels: pseudo-labels from TotalSegmentator were already sufficient to drive substantial gains, reinforcing the view that coarse anchors can be repurposed as powerful priors for downstream tasks.

In conclusion, AGC demonstrates that anatomical context offers a principled inductive bias for volumetric segmentation. By leveraging pseudo-label priors, encoding them as distance maps, and integrating them through feature-level modulation, AGC enhances boundary precision, reduces implausible predictions, and promotes anatomical coherence across diverse datasets and architectures.

## REPRODUCIBILITY STATEMENT

All datasets used in this study are publicly available: ImageCAS Zeng et al. (2023), PET/CT VAT, HAN-Seg Podobnik et al. (2023), and Tooth-Fairy2 Bolelli et al. (2024). Preprocessing details, model configurations, and training settings are described in Section 3 and Supplementary Notes 1–2. Ablation and comparison studies in Section 4 and Supplementary Note 3 further clarify design choices. To ensure reproducibility, we include the basic implementation code and the corresponding segmentation results (in `.nii.gz` format) within the supplementary zip file.

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

# A   APPENDIX

## USE OF LARGE LANGUAGE MODELS

During manuscript preparation, we used a large language model (ChatGPT) to assist in polishing English and improving readability. The model was employed exclusively for grammar correction, style refinement, and phrasing suggestions. All scientific content, experimental design, implementation, analysis, and conclusions are the work of the authors.

# Supplementary Materials

## Title: 3D Medical Image Segmentation with Anatomy-Guided Conditioning from Surrounding Structures

## B  SUPPLEMENTARY NOTE 1: DATASET-SPECIFIC ANATOMICAL PRIORS

We visualize the anatomical priors used to guide segmentation across seven tasks: **Coronary Artery**, **Visceral Fat**, **Carotid Artery**, **Mandible**, **Spinal Cord**, **Thyroid**, and **Alveolar Canal**. For each task, the semantic label map, HU intensity range, and 3D rendering of anatomical structures are provided. For ImageCAS and PET-CT datasets, priors were generated by TotalSegmentator, while HAN-Seg and Tooth-Fairy2 used original dataset labels. These priors are injected into the network through Anatomy-Guided Conditioning (AGC).

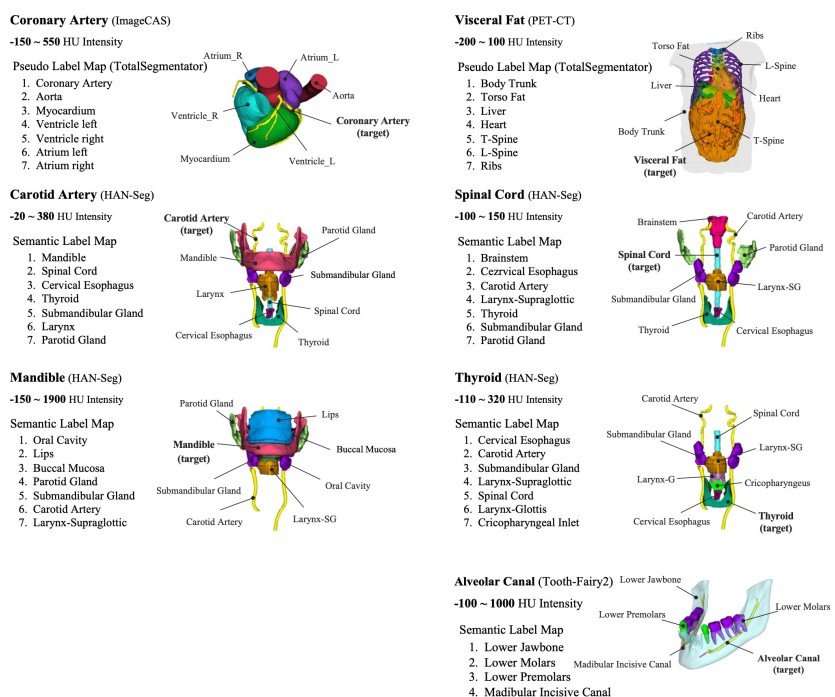

**Supplementary Figure 1:** Anatomical segmentation masks used for AGC across datasets. Target and contextual organs are shown in task-specific pastel colors, highlighting their spatial relation.

## C  SUPPLEMENTARY NOTE 2: MODEL- AND LOSS-AGNOSTIC EVALUATION

To evaluate generality, we provide two complementary analyses. **(i) Model-agnostic evaluation:** Supplementary Table 1 reports results on three anatomically distinct targets (VAT, Carotid, Tooth-Fairy2) using SegResNet, UNETR, and Swin-UNETR. **(ii) Loss-agnostic evaluation:** Supplementary Table 2 reports results with three losses (DiceFocal, SkelRec, clDice) on three datasets (ImageCAS, HAN-Seg, Tooth-Fairy2). In both cases, AGC improves overlap, boundary, and topology-aware metrics, with boldface marking improvements over baselines.

**Supplementary Table 1:** Model-agnostic evaluation of AGC on three datasets (VAT, Carotid, Tooth-Fairy2) with SegResNet, UNETR, and Swin-UNETR. Bold = improvement with AGC.

| Dataset | Backbone | AGC | DSC ↑ | HD95 ↓ | $\beta_0 \downarrow$ | $\beta_1 \downarrow$ |
|---|---|---|---|---|---|---|
| VAT (PET-CT) | SegResNet (Conv) | | 0.94 ± 0.02 | 1.82 ± 1.55 | 137.4 ± 73.5 | 129.1 ± 70.0 |
| | | ✓ | **0.96 ± 0.03** | **1.56 ± 1.05** | **128.6 ± 48.1** | **119.8 ± 44.6** |
| | UNETR (Transf.) | | 0.89 ± 0.04 | 38.3 ± 21.8 | 961.3 ± 509.5 | 927.9 ± 494.3 |
| | | ✓ | **0.95 ± 0.02** | **1.62 ± 1.19** | **159.1 ± 79.1** | **150.3 ± 74.1** |
| | Swin-UNETR (Hybrid) | | 0.88 ± 0.08 | 194.4 ± 27.6 | 977.5 ± 262.0 | 928.0 ± 246.3 |
| | | ✓ | **0.96 ± 0.02** | **1.75 ± 2.08** | **176.7 ± 79.1** | **167.7 ± 74.2** |
| Carotid (HAN-Seg) | SegResNet (Conv) | | 0.65 ± 0.07 | 269.6 ± 74.1 | 36.6 ± 20.2 | 33.3 ± 19.5 |
| | | ✓ | **0.73 ± 0.04** | **19.8 ± 5.15** | **6.25 ± 2.13** | **5.50 ± 1.94** |
| | UNETR (Transf.) | | 0.38 ± 0.06 | 437.1 ± 37.7 | 1618.3 ± 770.0 | 1459.8 ± 663.2 |
| | | ✓ | **0.55 ± 0.07** | **134.0 ± 164.7** | **78.9 ± 44.8** | **73.2 ± 42.2** |
| | Swin-UNETR (Hybrid) | | 0.27 ± 0.06 | 546.3 ± 38.9 | 561.5 ± 138.8 | 546.8 ± 137.8 |
| | | ✓ | **0.75 ± 0.05** | **27.0 ± 28.1** | **12.4 ± 5.53** | **11.2 ± 5.24** |
| Tooth-Fairy2 (Alveolar Canal) | SegResNet (Conv) | | 0.888 ± 0.016 | 2.56 ± 6.89 | 1.82 ± 1.41 | 1.64 ± 1.32 |
| | | ✓ | **0.982 ± 0.012** | **0.69 ± 0.77** | **0.45 ± 0.84** | **0.36 ± 0.69** |
| | UNETR (Transf.) | | 0.737 ± 0.076 | 22.0 ± 34.9 | 85.8 ± 22.6 | 82.6 ± 21.9 |
| | | ✓ | **0.966 ± 0.006** | **1.26 ± 1.07** | **3.88 ± 8.27** | **3.84 ± 8.22** |
| | Swin-UNETR (Hybrid) | | 0.889 ± 0.029 | 1.85 ± 1.63 | 1.43 ± 1.58 | 1.28 ± 1.47 |
| | | ✓ | **0.966 ± 0.005** | **1.01 ± 0.05** | **0.44 ± 0.75** | **0.42 ± 0.72** |

**Supplementary Table 2:** Loss-agnostic evaluation of AGC across three datasets (ImageCAS, HAN-Seg, Tooth-Fairy2). Bold = improvement with AGC.

| Dataset | Loss | AGC | DSC ↑ | HD95 ↓ | clDice ↑ | $\beta_0 \downarrow$ | $\beta_1 \downarrow$ |
|---|---|---|---|---|---|---|---|
| ImageCAS (Coronary) | DiceFocal | | 0.766 ± 0.032 | 81.8 ± 29.1 | 0.816 ± 0.077 | 25.2 ± 8.71 | 22.7 ± 8.30 |
| | | ✓ | **0.788 ± 0.021** | **36.4 ± 12.0** | **0.862 ± 0.054** | **10.6 ± 3.94** | **10.0 ± 3.99** |
| | SkelRec | | 0.778 ± 0.030 | 40.6 ± 17.8 | **0.856 ± 0.056** | 10.3 ± 5.13 | 9.77 ± 4.90 |
| | | ✓ | 0.778 ± 0.030 | **39.2 ± 14.7** | 0.845 ± 0.071 | **8.90 ± 3.89** | **8.32 ± 3.70** |
| | clDice | | 0.787 ± 0.033 | 27.9 ± 16.2 | 0.855 ± 0.057 | 18.1 ± 8.10 | 17.1 ± 7.77 |
| | | ✓ | **0.794 ± 0.023** | **27.2 ± 10.9** | **0.872 ± 0.051** | **7.19 ± 3.83** | **6.65 ± 3.58** |
| HAN-Seg (Carotid) | DiceFocal | | 0.646 ± 0.072 | 269.6 ± 74.1 | 0.646 ± 0.086 | 36.6 ± 20.2 | 33.3 ± 19.5 |
| | | ✓ | **0.727 ± 0.044** | **19.8 ± 5.15** | **0.789 ± 0.044** | **6.25 ± 2.13** | **5.50 ± 1.94** |
| | SkelRec | | 0.671 ± 0.060 | 108.2 ± 73.2 | 0.741 ± 0.105 | 36.5 ± 31.8 | 32.8 ± 26.4 |
| | | ✓ | **0.698 ± 0.062** | 111.5 ± 230.3 | **0.797 ± 0.192** | **14.9 ± 16.5** | **13.9 ± 15.8** |
| | clDice | | 0.671 ± 0.067 | 39.3 ± 31.6 | 0.801 ± 0.123 | 21.0 ± 20.2 | 17.8 ± 16.6 |
| | | ✓ | **0.673 ± 0.048** | **9.28 ± 4.87** | **0.842 ± 0.073** | **7.00 ± 4.43** | **6.00 ± 3.65** |
| Tooth-Fairy2 (Alveolar Canal) | DiceFocal | | 0.888 ± 0.016 | 2.56 ± 6.89 | 0.987 ± 0.018 | 1.82 ± 1.41 | 1.64 ± 1.32 |
| | | ✓ | **0.982 ± 0.012** | **0.69 ± 0.77** | **0.994 ± 0.009** | **0.45 ± 0.84** | **0.36 ± 0.69** |
| | SkelRec | | 0.880 ± 0.015 | 1.54 ± 0.32 | 0.994 ± 0.009 | 0.26 ± 0.48 | 0.24 ± 0.43 |
| | | ✓ | **0.988 ± 0.006** | **0.36 ± 0.60** | **0.998 ± 0.004** | **0.14 ± 0.35** | **0.12 ± 0.33** |
| | clDice | | 0.891 ± 0.012 | 1.48 ± 0.39 | 0.994 ± 0.011 | 0.24 ± 0.49 | 0.18 ± 0.45 |
| | | ✓ | **0.989 ± 0.006** | **0.31 ± 0.34** | **0.997 ± 0.005** | 0.24 ± 0.51 | 0.22 ± 0.50 |

# D SUPPLEMENTARY NOTE 3: ABLATION AND COMPARISON STUDY

Supplementary Table 3 quantitatively ablates AGC and largest-connected-component pruning (LCC), showing that AGC raises structural accuracy, LCC removes disconnected false positives, and the two combined deliver the best Dice and HD95.

**Supplementary Table 3:** Quantitative comparison of segmentation performance across four strategies: (1) no conditioning and no post-processing, (2) LCC post-processing only, (3) AGC only, and (4) AGC combined with LCC. All experiments were conducted using the SegResNet for clarity.

| Dataset | AGC | LCC | DSC ↑ | HD95 ↓ |
|---|---|---|---|---|
| Coronary Artery (ImageCAS) | | | 0.7658 ± 0.0321 | 81.8044 ± 29.0627 |
| | | ✓ | 0.7822 ± 0.0601 | 34.5192 ± 28.9655 |
| | ✓ | | 0.7883 ± 0.0208 | 36.4332 ± 12.0385 |
| | ✓ | ✓ | **0.7945 ± 0.0570** | **29.6120 ± 27.3706** |
| Visceral Adipose Fat (PET-CT VAT) | | | 0.9444 ± 0.0202 | 1.8175 ± 1.5481 |
| | | ✓ | 0.9400 ± 0.0392 | 3.4425 ± 9.8962 |
| | ✓ | | 0.9564 ± 0.0255 | **1.5603 ± 1.0483** |
| | ✓ | ✓ | **0.9565 ± 0.0267** | 1.7725 ± 2.2565 |
| Carotid Artery (HAN-Seg) | | | 0.6459 ± 0.0722 | 269.6247 ± 74.0853 |
| | | ✓ | 0.7276 ± 0.0616 | 26.8810 ± 9.1371 |
| | ✓ | | **0.7273 ± 0.0435** | **19.7951 ± 5.1479** |
| | ✓ | ✓ | 0.7035 ± 0.0745 | 34.1748 ± 16.6633 |
| Spinal Cord (HAN-Seg) | | | 0.7282 ± 0.0708 | 235.0251 ± 132.4872 |
| | | ✓ | **0.7617 ± 0.0362** | **5.8269 ± 3.7512** |
| | ✓ | | 0.7385 ± 0.0547 | 162.9246 ± 271.1081 |
| | ✓ | ✓ | 0.7503 ± 0.0403 | 6.4769 ± 3.9617 |
| Mandible (HAN-Seg) | | | 0.8446 ± 0.0393 | 261.8987 ± 31.0779 |
| | | ✓ | 0.9176 ± 0.0189 | 2.1617 ± 1.0773 |
| | ✓ | | 0.9150 ± 0.0198 | 3.0604 ± 4.1141 |
| | ✓ | ✓ | **0.9177 ± 0.0171** | **1.7538 ± 0.4173** |
| Thyroid (HAN-Seg) | | | 0.7782 ± 0.0925 | 215.6387 ± 129.9535 |
| | | ✓ | 0.8008 ± 0.0747 | 64.2508 ± 93.7234 |
| | ✓ | | 0.8131 ± 0.0811 | **6.5673 ± 4.6229** |
| | ✓ | ✓ | **0.8134 ± 0.0809** | 6.5946 ± 4.6689 |

# E SUPPLEMENTARY NOTE 4: ADDITIONAL RESULTS AND VISUALIZATION

We show additional qualitative examples that could not be accommodated in the main paper. Supplementary Figure 3 demonstrates that AGC consistently improves segmentation across two external datasets and several backbones. Supplementary Figures 4–6 then provide multi-subject 3D renderings for each benchmark—ImageCAS (coronary artery), PET-CT (visceral fat), and HAN-Seg (carotid artery, spinal cord, mandible, thyroid)—allowing visual inspection of common failure modes (fragmentation, boundary leakage, false positives) and their correction by AGC.

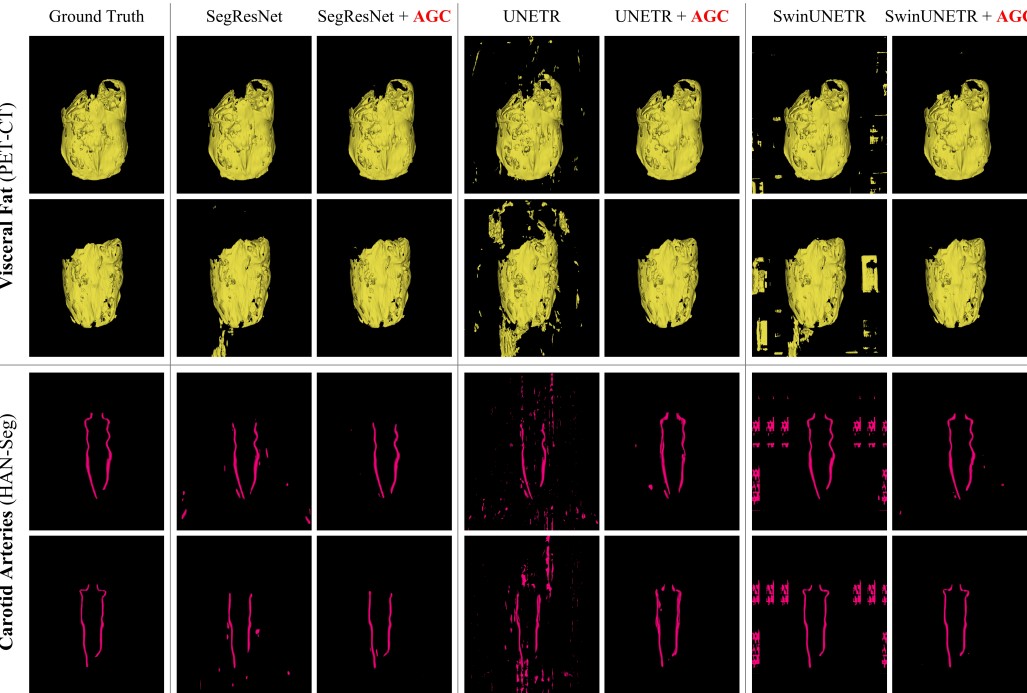

**Supplementary Figure 3:** Qualitative 3D visualization of segmentation results on two additional datasets, PET-CT (Visceral Fat) and HAN-Seg (Carotid Artery), using multiple segmentation backbones with and without AGC.

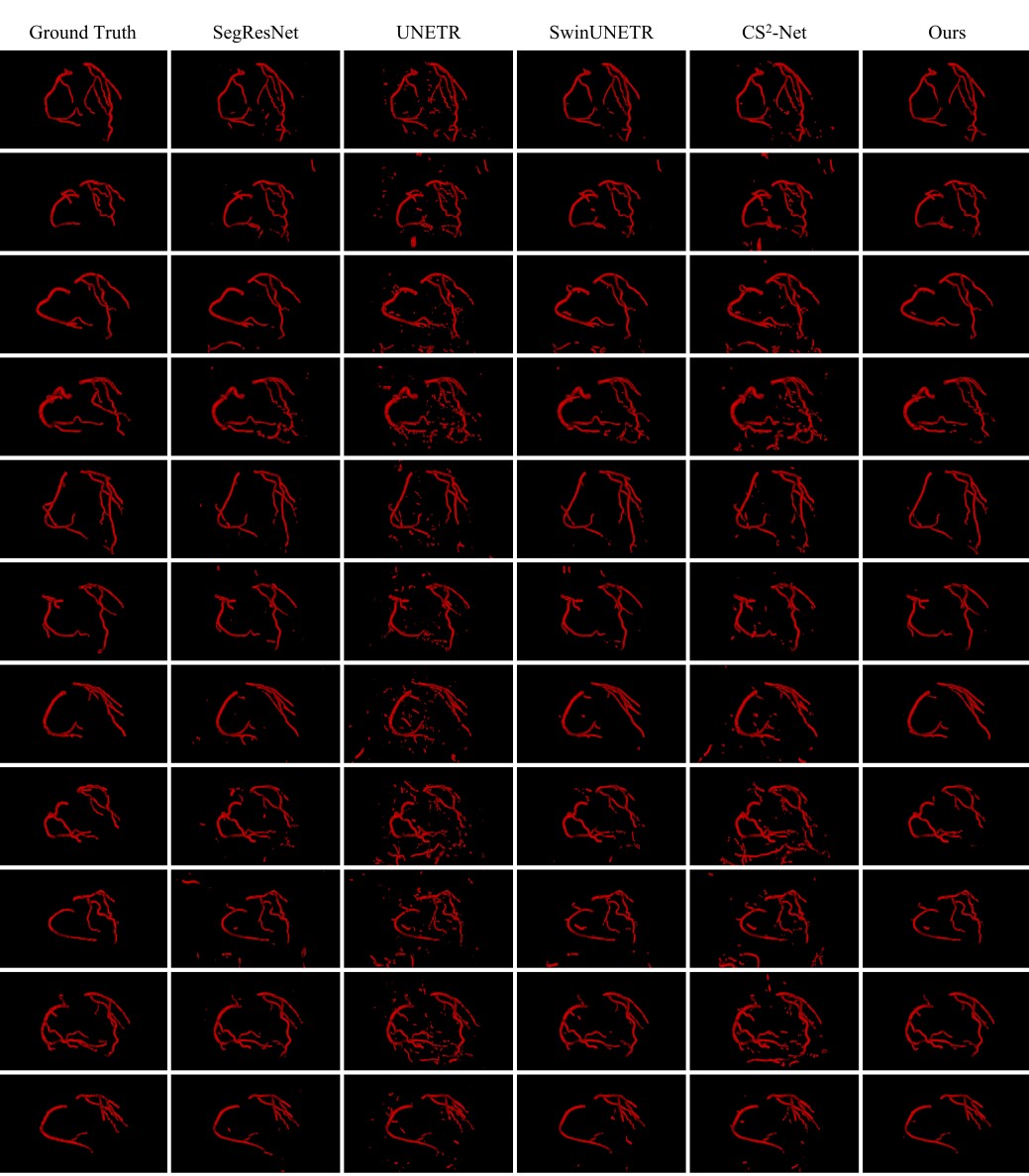

**Supplementary Figure 4:** Additional qualitative 3D visualization of coronary artery segmentation on multiple subjects from the ImageCAS dataset. Each row corresponds to a different subject, and each column shows the segmentation output from different segmentation backbones: Ground Truth, SegResNet, UNETR, SwinUNETR, CS$^2$-Net, and our proposed method (SegResNet + AGC).

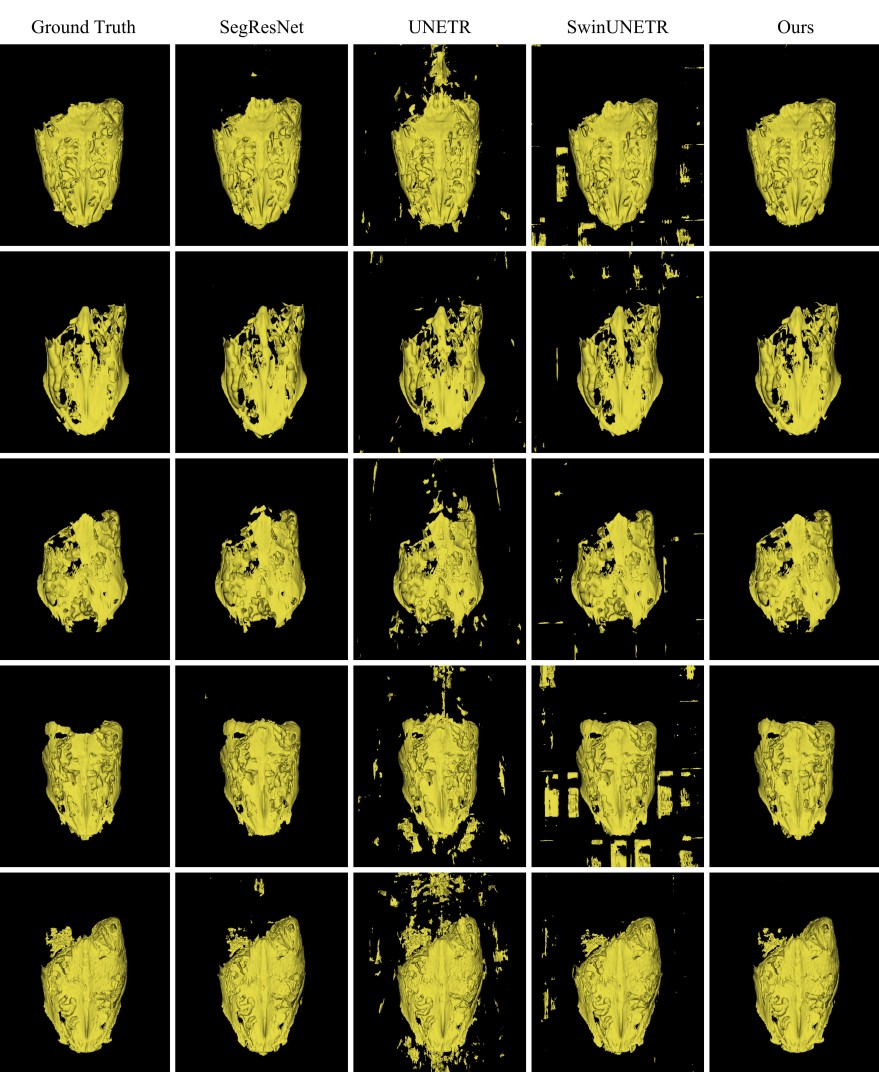

**Supplementary Figure 5:** Additional qualitative 3D visualization of visceral fat segmentation on multiple subjects from the PET-CT dataset. Each row corresponds to a different subject, and each column shows the segmentation output from different segmentation backbones: Ground Truth, Seg-ResNet, UNETR, SwinUNETR, and our proposed method (SegResNet + AGC).

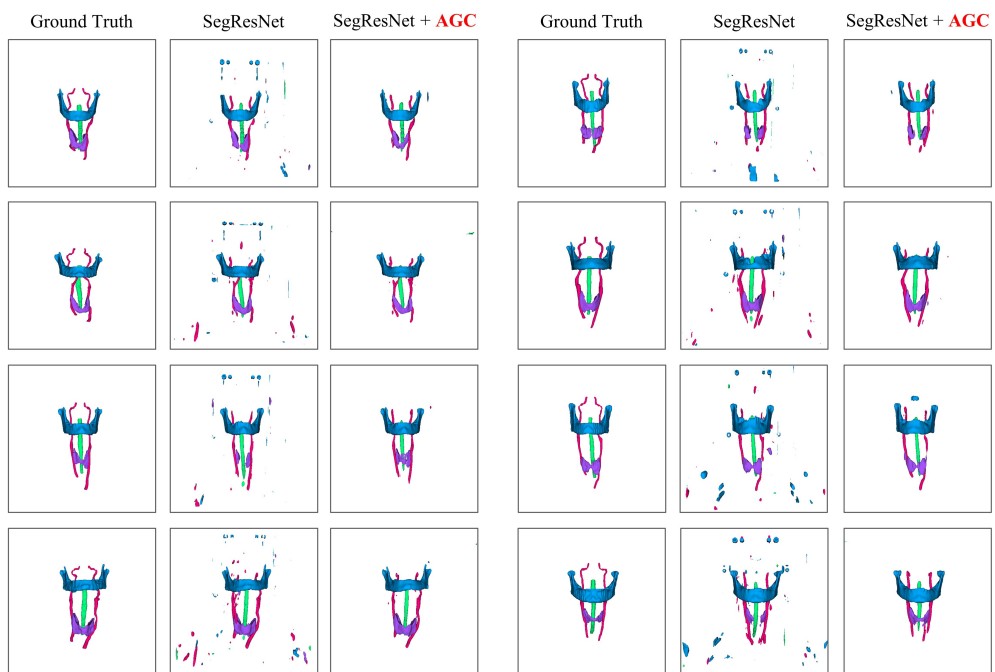

|  Ground Truth | SegResNet | SegResNet + **AGC** | Ground Truth | SegResNet | SegResNet + **AGC** |

**Supplementary Figure 6:** Additional 3D visualization results on the HAN-Seg dataset for multiple subjects. This figure presents the segmentation outputs of four anatomical structures—carotid artery, spinal cord, mandible, and thyroid. For each subject, we show the ground truth, SegResNet baseline, and the prediction refined by our proposed AGC method.

