# OpenReview forum: "3D Medical Image Segmentation with Anatomy-Guided Conditioning from Surrounding Structures"
_ICLR.cc/2026/Conference — ICLR 2026 Conference Withdrawn Submission_

### Official Review · Reviewer_jeTy · 2025-10-18

**Soundness:** 2
**Presentation:** 2
**Contribution:** 1
**Rating:** 2
**Confidence:** 5

**Summary:**

This paper proposes AGC for 3D medical image segmentation, which integrates signed distance maps of surrounding anatomy to introduce automatic priors that improve accuracy, boundary delineation, and topology without additional training. However, the practical utility of AGC and the completeness of the experiments both seem poor.

**Strengths:**

AGC introduces anatomical priors via signed distance maps, improving segmentation accuracy and topology-aware metrics on some existing models.

**Weaknesses:**

1. Lack of the sequence number of equations.
2. Lack of baseline evaluation with the latest approaches, such as CNN models nnUNet [1] and nnWNet [2], and SAM variant MedSAM2 [3] and H-SAM [4].
3. AGC relies on the full TotalSegmentator model to inject anatomical priors while introducing additional computational overhead and slowing inference efficiency. Stacking multiple models severely limits practical applicability; reporting parameters, FLOPs, and inference time is essential.
4. The datasets used, while suitable for assessing topological improvements, are small-scale and omit common 3D medical segmentation benchmarks such as BraTS2024 [5] and FLARE2022 [6].

Reference:

1. Isensee F, Jaeger P F, Kohl S A A, et al. nnU-Net: a self-configuring method for deep learning-based biomedical image segmentation[J]. Nature methods, 2021, 18(2): 203-211.

2. Zhou Y, Li L, Lu L, et al. nnWNet: Rethinking the Use of Transformers in Biomedical Image Segmentation and Calling for a Unified Evaluation Benchmark[C]//Proceedings of the Computer Vision and Pattern Recognition Conference. 2025: 20852-20862.

3. Ma J, Yang Z, Kim S, et al. Medsam2: Segment anything in 3d medical images and videos[J]. arXiv preprint arXiv:2504.03600, 2025.

4. Cheng Z, Wei Q, Zhu H, et al. Unleashing the potential of sam for medical adaptation via hierarchical decoding[C]//Proceedings of the IEEE/CVF conference on computer vision and pattern recognition. 2024: 3511-3522.

5. de Verdier M C, Saluja R, Gagnon L, et al. The 2024 brain tumor segmentation (brats) challenge: Glioma segmentation on post-treatment mri[J]. arXiv preprint arXiv:2405.18368, 2024.
6. Ma J, Zhang Y, Gu S, et al. Unleashing the strengths of unlabelled data in deep learning-assisted pan-cancer abdominal organ quantification: the FLARE22 challenge[J]. The Lancet Digital Health, 2024, 6(11): e815-e826.

**Questions:**

1. What does q denote in the equation (lines 173–176)?
2. Why are anatomical priors transformed into scale and shift parameters instead of using concatenation or other feature fusion strategies?

---

### Official Review · Reviewer_mMae · 2025-10-29

**Soundness:** 2
**Presentation:** 3
**Contribution:** 2
**Rating:** 6
**Confidence:** 3

**Summary:**

This paper proposes a method called Anatomy-Guided Conditioning (AGC), which integrates anatomical context from surrounding structures into segmentation networks. By leveraging signed distance maps of neighboring anatomy, AGC aims to enhance segmentation accuracy in 3D medical images. The method is applied to the decoder part of an encoder-decoder architecture, allowing for effective integration of anatomical priors without requiring significant modifications to the original network. The authors demonstrate AGC’s effectiveness across various datasets and different segmentation backbones.

**Strengths:**

1. The paper introduces a method for incorporating anatomical priors using signed distance maps, offering a clear advantage over traditional binary masks or coordinate embeddings.
2. AGC is shown to work effectively with a variety of encoder-decoder architectures without requiring modifications to the backbone, making it an easy-to-implement and scalable solution for many segmentation tasks.
3. The evaluation covers a wide range of datasets and diverse segmentation models, providing robust evidence of AGC's generalizability.

**Weaknesses:**

Lack of Clear Analysis of Computational Overhead.
The paper does not provide a detailed or clear analysis of the computational overhead introduced by AGC. While the proposed method is promising in terms of performance, it is important to understand the trade-off between its benefits and its impact on training and inference efficiency. Including this analysis would help assess the practical feasibility of the method in applications.

**Questions:**

1. For practical deployment, it is crucial to assess how AGC affects aspects like training speed, inference time, and memory consumption. The authors should include a more explicit discussion on these aspects and provide quantitative results in the form of tables comparing the computational cost (e.g., training time, inference time, memory usage) with and without AGC, as this would provide a clearer picture of the additional overhead introduced by the method.
2. Could the authors provide a more detailed breakdown of the impact of using signed distance maps versus binary masks or other prior conditioning methods? Does AGC offer a significant advantage over these alternatives? I may understand the rationality of using signed distance maps to represent anatomical structures, but I hope the author can provide a clearer explanation.

---

### Official Review · Reviewer_6LjU · 2025-11-12

**Soundness:** 3
**Presentation:** 3
**Contribution:** 2
**Rating:** 2
**Confidence:** 5

**Summary:**

This paper focused on a 3D medical image segmentation pipeline of thin, low-contrast, complex structures. The authors proposed an anatomy-guided conditioning module, which is an anatomically driven inductive bias for segmentation networks. The anatomical priors are extracted from the TotalSegmentator dataset, and encoded as signed distance maps (SDMs), then it is fed into a feature moduleation block in the decoder for segmentation prediction.

**Strengths:**

- The paper utilized a pre-defined comprehensive dataset for extracting the anatomical priors and used it for further processing.
- The method is evaluated in many complex structure datasets, such as arteries, visceral adipose tissue, head and neck organs, etc.

**Weaknesses:**

- The anatomically guided methods are widely adopted for medical image segmentation methods, including hierarchical methods, topological methods, etc. The work shall clarify why this anatomically guided conditioning method is superior, and more ablative studies may be needed to explain its effectiveness.
- Typically, if the method is designed with structure-aware metrics, it won’t perform well in general anatomies. How do the methods perform directly on the total segmentator dataset or other whole-body CT datasets?
- The TotalSegmentator dataset is mainly generated from nnUNET, not purely from human annotation. I understand this work’s assumption is based on priors from TotalSegmentator, but it might create a circular proven issue for gold standards. The learned priors is also from AI models.
- Baseline UNETR shall also be a hybrid model, same as Swin UNETR and nnFormer, a minor issue, but it’s good to clarify.

**Questions:**

Question, see weaknesses section.

---

### Note · Authors · 2025-11-12

I have read and agree with the venue's withdrawal policy on behalf of myself and my co-authors.